# Reducing the Distortion in Particle Filled Material Extrusion (MEX)-Based Additive Manufacturing (AM) by Means of Modifying the Printing Strategy

Johannes Abel [1,*], Siddharth Tiwari [2], Milán Kardos [2], Maria Reichel [1] and Uwe Scheithauer [1]

[1] Fraunhofer Institute for Ceramic Technologies and Systems IKTS, Winterbergstraße 28, 01277 Dresden, Germany

[2] 3D Ceram Sinto Tiwari GmbH, Rudower Chaussee 29, 12489 Berlin, Germany

* Correspondence: johannes.abel@ikts.fraunhofer.de; Tel.: +49-351-2553-7502

**Abstract:** This study addresses a ubiquitous challenge in powder metallurgy: sintering distortion. Sintering distortion can have various causes. On one hand, external factors such as friction with the sintering support during sintering or temperature gradients in the furnace, and, on the other hand, internal factors such as anisotropic shrinkage due to directional layer build-up or residual stresses during production, can cause deformation by relieving mechanical stress. This paper presents an approach to reducing residual stresses in components produced by ceramic Fused Filament Fabrication (CerAM FFF) by changing the printing strategy using thermoplastic porcelain filaments with a solid loading of 57% vol. The starting point of the investigation was the torsion of standard sliced porcelain fragments after solvent debinding, which led to the idea to change the printing direction to prevent the distortion. Therefore, a Python™-based post-processor was developed to control the printing direction. It has been shown that this approach can even prevent warpage both for printed ceramic and also for the metal components for technical applications. This simple observation will help all powder metallurgical manufacturers using Material Extrusion (MEX)-based Additive Manufacturing (AM).

**Keywords:** fused filament fabrication; additive manufacturing; CerAMfacturing; ceramic; metal; distortion; porcelain; material extrusion; 3D printing

## 1. Introduction

Additive Manufacturing (AM) is finding its way into almost all industries. Not only in the technical but also in the field of arts, there are applications where either the complexity or the small number of pieces justifies the use of such processes. One interesting application is the restoration of valuable porcelain art objects that have, for example, been damaged during or in connection with the 2nd World War. As the fracture of porcelain is usually random, no two pieces are alike. Rapid prototyping processes can help to manufacture individual replacement parts. In one project, broken vases from the famous Meissner porcelain collection of August the Strong, in Dresden Germany, are taken into focus to be restored. In particular, a splendid vase made by Joachim Kändler in the 18th century was identified to be supplemented by a spare part made by Ceramic Additive Manufacturing (CerAM). Figure 1, on the left-hand side, shows the damaged vase and the second still, on the right-hand side, shows the intact vase where the missing part was scanned and reconstructed (Figure 1 right).

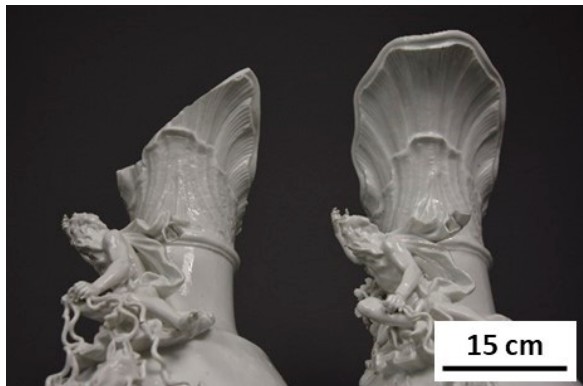 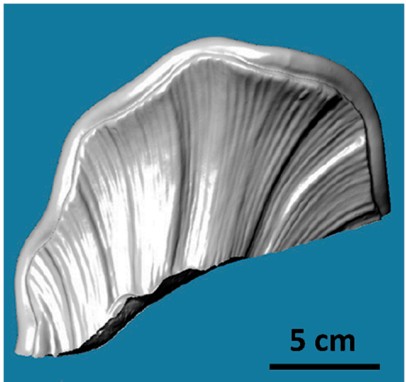

**Figure 1.** (**Left**): Splendid vases made by Johann Joachim Kändler in the 18th century (courtesy: Staatliche Porzellansammlung Dresden, Germany); (**right**): 3D-reconstruction of the missing fragment with fracture surface (courtesy: Cox3D Pirna, Germany).

Most of the replacements were made of polymers and are mostly manufactured by hand. Unfortunately, these compounds have a color change over the years due to exposure to light. To overcome this, the original Meissner Porcelain was utilized for additive manufacturing of the missing fragment, maintaining its original color and overall appearance. This project, entitled "RestaurAM", was funded by the German Zentrales Innovations program Mittelstand (ZIM), with the grant no. ZF4076454AG9. As the fragment is a narrow but large component (high aspect ratio), the probability of distortion during the treatment after the green part manufacturing is high. If the sprout deforms too much, it will never fit the counter fracture surface of the splendid vase.

For the manufacturing of the parts, Fused Filament Fabrication (FFF), which belongs to the class of Material Extrusion (MEX) technologies, was used to manufacture the porcelain fragments. Particular advantages of the FFF process are the comparatively low machine acquisition costs, the component size scalability, the multi-material capability and the wide powder and particle size spectrum. Its disadvantages are the comparatively low resolution, which results in rough (rippled) surfaces or staircase-like features and reduced geometric tolerances compared to, e.g., stereolithographic processes such as Vat Photopolymerization (VPP) [1]. A well elaborated overview of MEX is given by Gonzales-Gutierrez et. al. and is considered to be state of the art [2].

For FFF, self-made filaments with a solid loading of 57% vol. were used. After the mandatory solvent pre-debinding, it turned out that a characteristic distortion of the fragment took place, which remained even after sintering, preventing a form fit to the counterpart. Due to the characteristic distortion, it was evident that a systematic reason was responsible. Therefore, it was assumed that the printing strategy had a strong influence. To investigate the certain distortion directions, simple squares manufactured with different printing directions were processed and characterized.

In this paper, the layer-wise printing direction is changed to reduce the part distortion during the debinding of porcelain components by post-processing the G-code. In addition, it is shown that this approach is also effective for alumina ($Al_2O_3$), 17–4 PH and the titanium alloy (Ti6Al4V). Through the presented study, it is successfully demonstrated that this elaborated procedure is a valuable slicing tool for highly particle-filled, low-distortion MEX-based AM, in all fields of application.

## 1.1. Part Distortion during Processing

Distortion (warpage, deformation) is one of the major issues in powder metallurgy, particularly when polymers have a high proportion of the material in the feedstock. Distortion is always related to mechanical stresses. Mechanical stresses during part manufacturing can be introduced by, e.g., the applied pressure or temperature history during shaping, or temperature gradients during the heat treatment [3]. If the stresses are critical, defects

such as cracks occur. During relaxation, stresses are reduced with increasing deformation. The degree of storing and the reduction in mechanical stresses in thermoplastic materials depends on their viscoelastic properties. These are described in the theories of the complex shear modulus $G^*$ [4].

$$G^* = G' + iG'',$$ (1)

Following (1), $G^*$ contains the storage modulus $G'$ and the loss modulus $G''$. Ideal elastic materials exhibit $G'' = 0$ (Hook's law), whereby ideal viscous materials exhibit $G' = 0$. Thermoplastic materials usually exhibit $G''$ and $G' \neq 0$ related to their molecule orientation and are therefore called viscoelastic. Deeper insights into the theories are described elsewhere [4,5]. In addition to the flow-induced stresses, thermal-induced stresses also play an important role. These contain rheological and thermophysical parameters [3].

For now, it is sufficient to understand that stresses can be dissipated or stored, causing deformation or relocation of material above the materials yield strength. The yield strength from a certain thermoplastic material is dependent on the thermophysical properties, state of isotropy and stresses at a certain temperature.

This paper will not attempt to elucidate the contribution of the FFF process to viscoelastic, pressure or temperature-induced stresses, but is intended to increase awareness of the physical parameters that must be included in such considerations to prevent distortion. An appropriately adapted slicing strategy is demonstrated.

### 1.2. Development of Stresses in the Applied Process

The additive manufacturing method chosen here is the Fused Filament Fabrication-method of ceramics (CerAM FFF). Following the DIN EN ISO/ASTM 52,900, this method belongs to the material extrusion techniques (MEX). A thermoplastic, wire-like, semifinished product (filament) is molten in a heated nozzle and solidified during cooling below it. During the process, a continuous strand is extruded, with a speed of typically 5 mm/s to 60 mm/s, through the nozzle on top of the previous layer. The melt is typically deflected by 90° under the hot nozzle, which is positioned vertically on the component with a distance of one layer height, which is typically between 50 μm and 300 μm. Due to this relative motion, it is assumed that the strand is tensioned—i.e., stress is introduced—if the printhead speed exceeds the material exit speed. The resultant stress can be imagined to be similar to a helical cylindric spring that is being continuously twisted around the rotational axis as the object is manufactured. It is assumed that these stresses are relaxed during the subsequent processing and therefore cause the described phenomenon of deformation. Stresses are relaxed until they reach the yield strength. As the yield strength is dependent on the viscosity, which in turn is dependent on temperature at very low shear rates, the deformation can take place in the treatments after green body manufacturing, where these aspects are varying. These are the debinding processes such as solvent pre-debinding, thermal debinding and sintering. As presented in the literature, the most fragile state is reached after the complete thermal removal of the binder in the so-called brown state, when the particles are only in frictional contact [6]. As the ceramic particles are elastic and low bonded, stresses are often critical, for example, leading to destroying the part during handling. The remaining residual stresses in the brown part can lead to deformation or even cracking during sintering where the viscosity and the transition between plastic flow and elastic deformation is dependent on temperature.

A process often applied prior to thermal debinding is chemical pre-debinding via solvent extraction processes. In this process, a certain amount of thermoplastic binder is removed from the additive manufactured body (green body). Most of the thermoplastic binders consist of multiple organic constituents. such as wetting agents, lubricants and back bone polymers, and are therefore called binder systems. Typically, higher molecular polymers remain in the body after solvent debinding, acting as back bone polymer, giving the body a certain strength to allow, for example, the transfer to a furnace for subsequent thermal debinding. Furthermore, binder systems provide a staggered degradation of constituents during thermal debinding, preventing an excessive outgassing and, consequently,

a pressure increase in the component. Usually, the lower molecular polymers, such as lubricants (e.g., waxes), are extracted during solvent debinding, beginning from the outside to inside. The reduction in the total organic content by such a solvent pre-debinding has advantages in terms of the subsequent thermal debinding, reducing the amount of binder that needs to debinded thermally and, subsequently, the time required for thermal debinding. In thermal debinding, the polymers are cracked and degraded by increasing their specific volume due to the phase change from solid to gas. As the powder metallurgical feedstocks are highly solid-loaded, the outgassing and the diffusion of the gas is limited by the pore sizes. Consequently, a pressure within the component is formed, which can lead to blown/extended pore channels and even to blistering, deformation or critical defects [7]. In principle, a high amount of extracted binder is desirable and homogeneously dissolved from shell to core, which is also reported in the literature [8–10].

Another effect that can often be observed is the swelling of polymers in the solvent as the solvent diffuses into the minor binder [10,11]. Swelling of the polymer is caused by introducing inhomogeneous stresses as the immersion of the solvent takes place from the outside to the inside of the component. Therefore, it is evident that these stresses are in superposition to those mentioned above.

To conclude, it is important to note that debinding is the most critical step of powder metallurgical part manufacturing regarding distortion and the formation of defects.

### 1.3. State of the Art: Distortion of Thermoplastic Components

Kamal et.al. [5] investigated residual stresses in injection molded polystyrene and high-density polyethylene samples. They found that, depending on the molding conditions and wall thickness, residual stresses in the range of 5 MPa and 10 MPa remain in the plate-like samples. These values give at least an idea of the magnitude of stress. Nevertheless, the remaining stress values can differ depending on the material and processes. Jansen et. al. [12] also report stresses in freely quenched molded parts in the same range of magnitude, using general analytical expressions including effects of pressure, external forces, crystallization and reaction.

Zhang et al. [13] used a finite element analysis model and measured prototypes to study the effect of different tool path patterns on residual stresses and distortions on 3D-Printed parts with FFF (Fused Filament Fabrication). Although the cooling effects of different infill directions were investigated, it was proven that changing the tool path directions does influence the internal stresses in the part.

Samy et al. [14] investigated the influence of raster patterns on residual stress and distortion using semi crystalline polymers such as polypropylene (PP). They found that the kind of raster pattern and its deposition direction influence the distortion, as thermal effects such as phase transition (amorphous to crystalline) result in density gradients, leading to stresses in multilayers.

## 2. Materials and Methods

To investigate the systematic appearance of distortion, it is evident that simple geometries must be manufactured to derive the main effects of the applied processing. Therefore, simple box squares were manufactured and processed prior to the manufacturing of the complex fragments (Figure 1). The filament was produced at Fraunhofer IKTS, utilizing a kneader and twin-screw extrusion and followed by the spooling procedure. The filament was processed in a slightly modified FFF Prusa i3 MK3 S+ (Prusa Research, Prague, Czech Republic) printer using an ordinary brass nozzle. The modification of the printer included the cooling of the filament prior to being extruded to avoid softening during the filament conveying process towards the hot nozzle. The early softening can cause blocking for ceramic filaments due to their low melting point compared to thermoplastic filaments such as PLA, ABS, etc. The wet milled and dried porcelain powder was supplied by KI Keramik-Institut GmbH Meissen, Germany, based on the traditional porcelain recipe with a $d_{50}$ of 1.65 µm. The thermoplastic non-commercial polyamide-like binder system was

provided by Inmatec Technologies GmbH, Reihnbach, Germany. To investigate the porosity, the Archimedes Method was used, in which the samples were immersed in distilled water. An overview of the process chain is given in Figure 2.

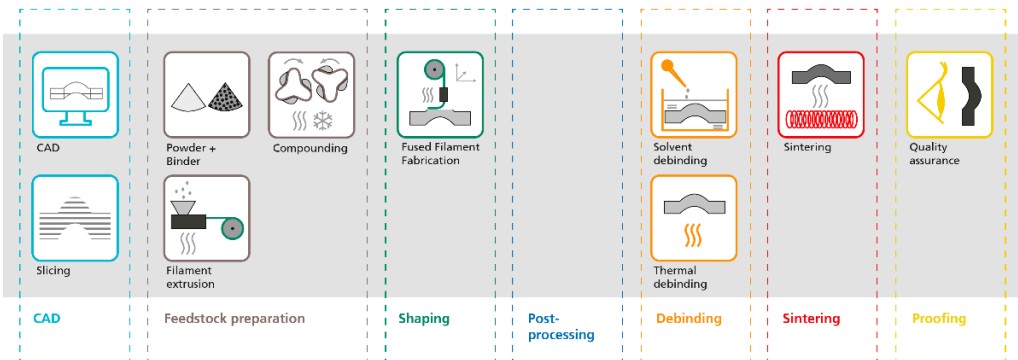

**Figure 2.** Overview about the process chain of CerAM FFF.

### 2.1. CAD Modelling

After the light-based scanning and remodeling of the fragment, the CAD-Data from the fragment was supplied by COX3D Pirna, Germany. The 3D data is exported as STL-file (Standard Tessellation Language). This file is then post-processed to achieve the different printing strategies elaborated in Sections 2.2 and 2.3.

### 2.2. G-Code

The G-Codes were generated using Ultimaker Cura 4.13.1 (Ultimaker, Utrecht, The Netherlands) open-source slicing software. The G-Code commands are executed by the 3D-Printer to create the green body by moving stepper motors responsible for positioning the nozzle, heating and extruding the material. Most of the lines in the code are commands, which are interpreted by the 3D printer's firmware to carry out movements or set temperatures. An example of a "G1" command, which realizes a feed-rate setting, movements, and extrusion is:

<p align="center">G1 F500 X10 Y0 E0.5.</p>

When this command is run, the toolhead travels to the X = 10 [mm]; Y = 0 [mm] coordinates in a straight line. This is conducted with a maximum feed-rate of 500 mm/min, while extruding 0.5 units of filament, assuming the last "E" command was 0. This means that the amount of extrusion defined by the "E" command is determined by the difference between the previously set "E" value and the new one found in the command. Movements involving arcs are realized by multiple straight commands in high resolution.

Other commands in the G-Code generated by the slicer software include comments indicating layer numbers and the type of lines being printed. This makes it more transparent and helps with navigation in the G-Code.

### 2.3. G-Code Post Processing

The post-processing script written in Python™ 3.8 (Python Software Foundation, Wilmington, United States of America) programming language was implemented in the slicer software, and its execution when saving the file could be enabled or disabled inside Cura. This means that the G-Code modification was seemingly integrated into the slicing step in the process chain.

The post-processing script changes the perimeter directions in every $n^{th}$ layer from clockwise (CW) to counterclockwise (CCW), or vice versa, where the number n can be set in the slicer software. The effect is represented in Figure 3, where n would be 2.

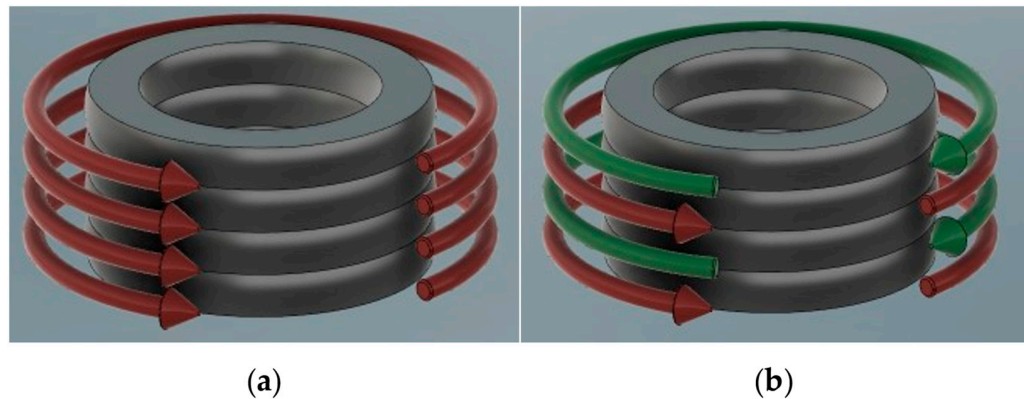

**Figure 3.** Goal of the G-Code post-processing. (**a**): Original perimeter directions (counterclockwise); (**b**): Perimeter directions after post-processing (alternating directions).

Iterating through each line of the code and using the information of the comments generated by the slicer software, it is possible to determine the purpose of each command. The script keeps track of layer numbers and the type of lines being printed. The line type printed by the G1 commands could be a perimeter, infill, top surface, etc. With this method, whenever a perimeter that must be reversed is detected, it is also being stored while iterating through its commands. Once a whole perimeter-section is stored, the post-processing algorithm transforms and replaces it.

The main principle of this algorithm is demonstrated in Figure 4 and Table 1.

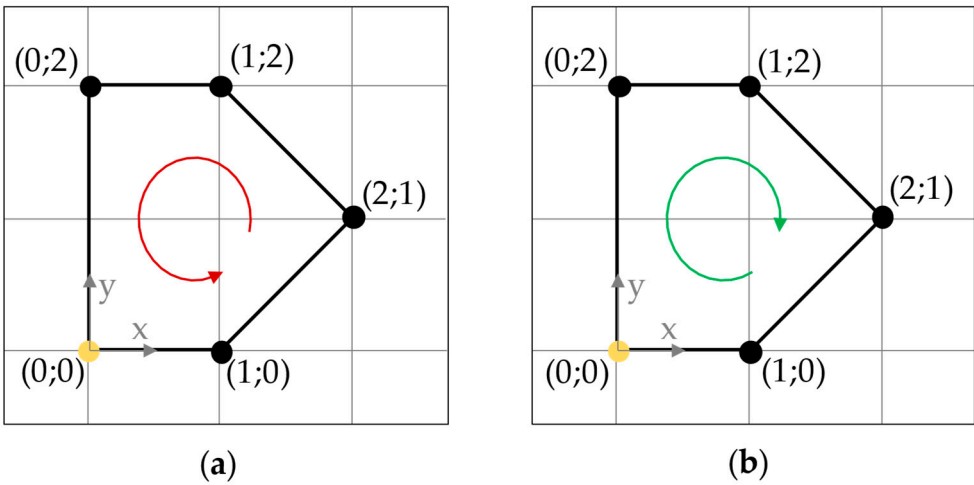

**Figure 4.** Example of perimeter direction change; (**a**): CCW, (**b**): CW; (X;Y).

**Table 1.** Example of perimeter direction change.

| Line Nr. | Original G-Code | Reversed Direction |
|:---:|:---:|:---:|
| 1 | G1 F500 X1 Y0 E1 | G1 F500 X0 Y2 E2 |
| 2 | G1 X2 Y1 E2.4 | G1 X1 Y2 E3 |
| 3 | G1 X1 Y2 E3.8 | G1 X2 Y1 E4.4 |
| 4 | G1 X0 Y2 E4.8 | G1 X1 Y0 E5.8 |
| 5 | G1 X0 Y0 E6.8 | G1 X0 Y0 E6.8 |

Regarding the X and Y coordinates, the order of commands is simply reversed and then the first command is cut and pasted to the last place. This way, the direction of the shape is reversed and the toolhead ends up at the starting point, as intended.

Care must be taken when calculating the extrusion (E) values, as they increase with each command and cannot be simply reversed or reassigned to other coordinates. This

means that each G1 travel might have its specific change in the E value from the previous one, depending on the length extruded. These differences are calculated for each command, summed up in reverse order and reassigned to the correct G1 command.

The value of feed-rate (F) is always set at the beginning of the perimeter section, and simply reorganizing the command would mean that an old F value would be used. This means that the feed-rate value is also added to the "new" first command by the script.

A similar approach was recently implemented in the Ultimaker Cura 5.0 beta (Ultimaker, Utrecht, The Netherlands) with an Arachne engine for alternate extra wall to reduce the distortion. No investigations into reducing the part deformation using ceramic materials had been found in the literature so far.

### 2.4. Filament Preparation

The pre-mixing of the 57 vol.-% porcelain containing thermoplastic material was performed using a heated double z-kneader LKII (Hermann Linden Maschinenfabrik GmbH & Co. KG, Marienheide, Germany) at a temperature of approximately 80 °C, aiming for a homogeneous mixture without a complete melting of the binder.

The pre-mix was fed into a twin-screw extruder KETSE 20/40 (Brabender, Duisburg, Germany) to fully homogenize and granulate the material. After two throughputs, the pressure in the extruder was constant, indicating a sufficient mixed material ready for continuous filament production. Therefore, the cylindrical granulate was put into the twin-screw extruder another time and was extruded at a 130 °C nozzle temperature using a round shaped nozzle, 1.9 mm in diameter, into a continuous filament. This filament was cooled down on a conveyor belt, measured by a line laser and spooled on a standard spool. By controlling the pull-off speed, the diameter can be varied in small ranges to produce round filaments with a diameter of ~1.75 mm.

### 2.5. Shaping: FFF Process

For the FFF process, the following parameters (Table 2) had been applied for the square boxes (35 mm x 35 mm x 20 mm (l x w x h)) and the scaled (0.85) fragment (Figure 1 right). For manufacturing reasons, the fracture surface had been cut off the fragment maintaining an even built up. The fragments had four perimeters and a 20% infill pattern of the type $+45°/-45°$. The squares consist of parallel perimeters only.

**Table 2.** Used parameters for FFF.

| Parameter | Value |
| --- | --- |
| nozzle diameter (mm) | 0.4 |
| temperature (°C) | 155 |
| layer height (μm) | 150 |
| speed (mm/s) | 30 |
| temperature building platform | room temperature |

According to the consideration that the printing direction (movement of the print head in xy-plane) has an impact on the distortion, the samples were manufactured as shown in the following Table 3.

**Table 3.** Manufacturing properties of the samples.

| | Direction | Perimeter | Infill (%, Type, Direction) |
| --- | --- | --- | --- |
| square 1 | clockwise | 8 lines | - |
| square 2 | counterclockwise | 8 lines | - |
| square 3 | alternating (*n* = 2) | 8 lines | - |
| fragment 1 | clockwise | 4 lines | 20%, $+45°/-45°$, default |
| fragment 3 | alternating (*n* = 2) | 4 lines | 20%, $+45°/-45°$, default |

## 2.6. Debinding and SINTERING

The debinding was carried out by a two-step process. First, a solvent debinding in acetone was applied over 48 h, at a solvent temperature of 35 °C and using a 30-L debinding device MDU30 (DesbaTec Anlagentechnik GmbH, Sulzbach, Germany). After a slow drying, the samples were thermally debindered in a muffle kiln L 15/11 (Nabertherm GmbH, Lilienthal, Germany) with a heating rate of 0.5 K/min up to 600 °C in air. The complete debinded samples were subsequently sintered with a heating rate of 3 K/min up to 1250 °C and a 15 min dwell time in air utilizing the kiln HT 08/18 (Nabertherm GmbH, Lilienthal, Germany). The aqueous glaze suspension from KI Keramik-Institut GmbH Meissen, Germany, was applied on the pre-sintered body (1000 °C) by spraying and was fired at 1250 °C in air.

## 2.7. Proofing

The distortion of the squares was measured by means of an optical stripe light scanning system ATOS Core 45 (Carl Zeiss GOM Metrology GmbH, Germany). The 3-dimensional measurement of the fragments was conducted by computed tomography (CT), utilizing a CT Compact (Procon X-ray, Sarstedt, Germany) with an applied voxel size of 63 μm. The samples were scanned with 130 kV and 124 μA, at an exposure time of 405 ms.

## 3. Results and Discussion

The produced homogeneous filament (Figure 5) exhibits superior flexibility compared to other ceramic filaments on the market and, at the same time, possesses rigid properties to be processed at standard printers. The bending radius of the filament was ~10 mm before breaking. The reached diameter of the filament was 1.76 mm (±0.02 mm) and fulfils all requirements for good green part manufacturing. The layers are very fine and smooth, as seen in Figure 5.

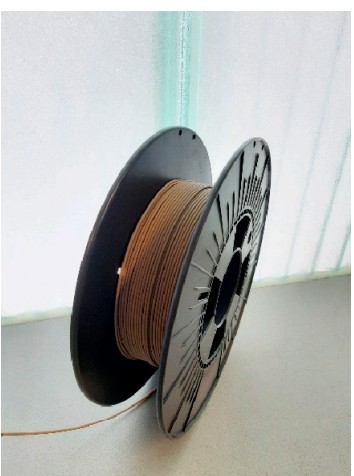

**Figure 5.** Self-made and spooled Meissner porcelain filament ø = 1.75 mm.

For the animation of the G-code, the PrusaSlicer G-code Viewer- 2.5.0 (Prusa Research, Prague, Czech Republic) was utilized. In order to investigate the distortion, the square box green parts consisting of eight parallel lines (Figure 6) were 3D measured after solvent debinding and sintering. Thereafter, the manufacturing directions: clockwise, counterclockwise and alternating (*n* = 2) were investigated.

Two planes in different z-heights were considered (Figure 7 left). The two contours were then superimposed normally to each other to visualize the twisting of the square contour (Figure 7 right). Only one representative corner of the square was analyzed in the following, Figures 7–9.

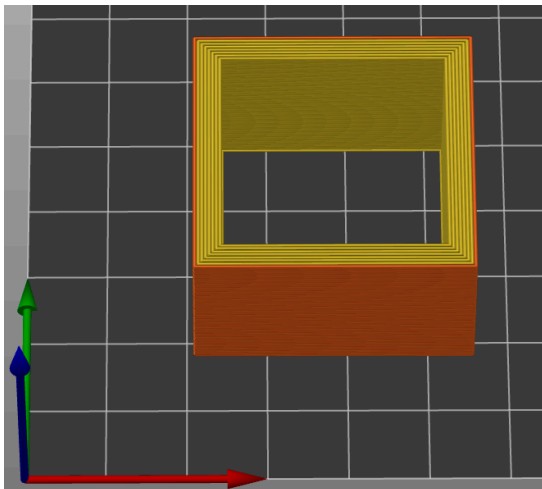

**Figure 6.** Animated G-code of the square box green part (35 mm × 35 mm × 20 mm (l × w × h)).

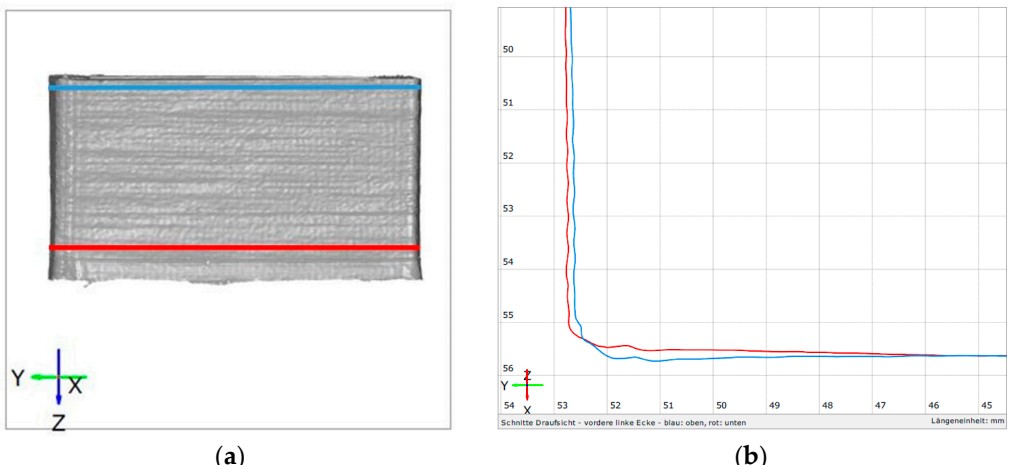

(**a**)                                                                                        (**b**)

**Figure 7.** GOM measurement of square 1; (**a**): two investigated planes (blue and red); (**b**): superimposed planes from top view; manufacturing direction: clockwise; solvent debinded.

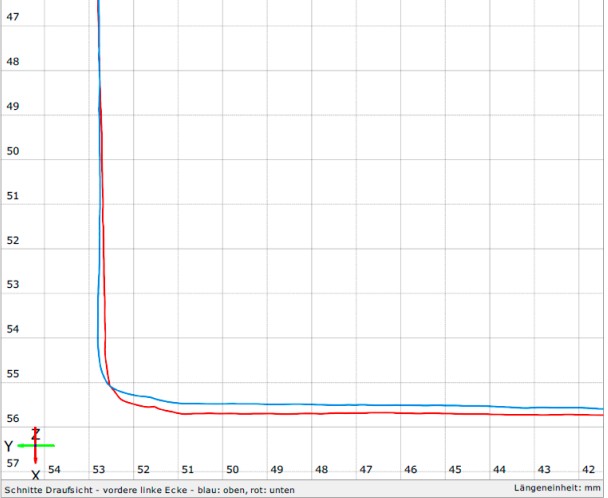

**Figure 8.** GOM measurement of superimposed planes from top view; manufacturing direction: counterclockwise; solvent debinded.

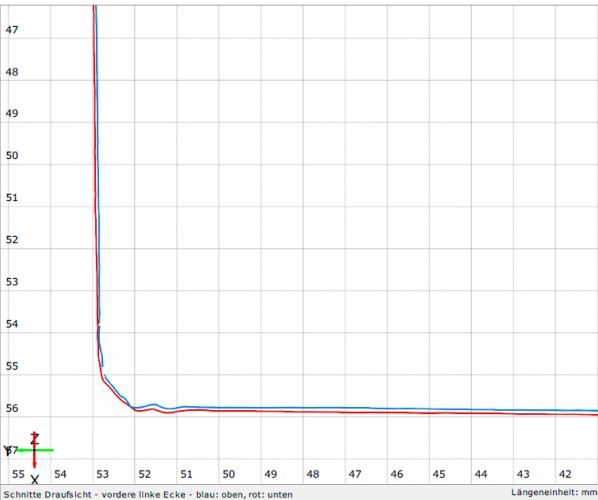

**Figure 9.** GOM measurement of superimposed planes from top view; manufacturing direction: alternating (*n* = 2); solvent debinded.

It turned out that the blue and red contour are not congruent in top view, which indicates a twisting of the structure during solvent debinding. The deformation takes place counterclockwise, if the bottom line (red) is assumed to be the fixed base line. If the manufacturing direction is counterclockwise, the deformation is clockwise, as shown in Figure 8. The deformations can be seen with the naked eye. An impressive picture is revealed when looking at Figure 9. No significant deformation is detected in this case. It is assumed that the stresses leading to deformation, while manufacturing in the same direction is compensated by such a change in direction. The deformation was not found in the green state, which led to the assumption that a softening took place during the solvent debinding, releasing stresses by increasing the deformation.

After sintering the samples, the same qualitative appearance of distortion, with no significant change, was observed. The twisted samples remained twisted, the alternating one stayed accurate. It is assumed that, with this observation, larger parts such as the fragments could be manufactured with less distortion, necessary for form-fitting to the fracture surface of the splendid vase. Therefore, the G-codes of the fragment was post-processed with *n* = 2. An animated G-code of a cut fragment is shown in Figure 10. Figure 11 shows a green part without the bottom fracture surface. To investigate the distortion of the fragment, a comparison between the CAD-model and the green state printed clockwise (fragment 1) or alternating with *n* = 2 (fragment 3), as well as the corresponding solvent debinded and sintered component to the CAD-Model, was made. To scan the manufactured component in each state, the computer tomograph CT compact was utilized. A superimposed visualization of the CAD-Model (blue) and the green part (gray) is illustrated in Figure 12. The penetrating colors show slight deviations of the manufactured part from the CAD-model. To quantify the deviations, the normalized distance (best fit) of discrete points are illustrated in Figure 13. The green color indicates a good conformity in the chosen value range of ±4.5 mm.

The deviations of the surface are in the range of ± 0.5 mm. These deviations are caused by the printing and matching issues by the comparison of the CT measurement and CAD-data. Almost the same deviations from the green state to the CAD-model occur if the alternating manufactured fragment 3 is investigated in the same way.

After solvent debinding, a completely different deformation occurred, as suspected, following the observations of the squares above. The weight loss after solvent debinding was calculated to 18 wt.-%. This corresponds to ~71% of the total amount of binder in the filament. The best-fit deviations of the clockwise-manufactured component, compared to the CAD-model, grew drastically, up to ± 4.5 mm, as shown in Figure 14. In the lower left corner of the component, a twisting of the baseplate can be observed (yellow to blue =

sign change), which leads to the fact that such a fragment would not be able to accurately reproduce the fracture surface.

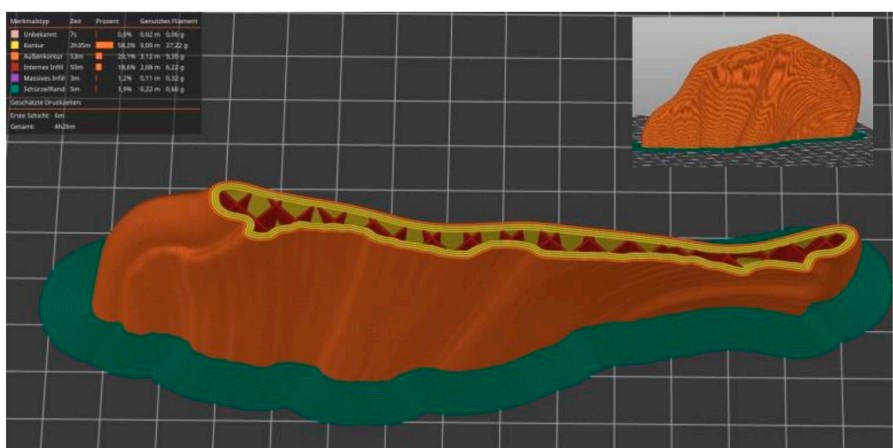

**Figure 10.** Cut and downscaled fragment animated G-Code (4 perimeters, Infill: 20%, +45°/−45°, infill manufacturing direction: default).

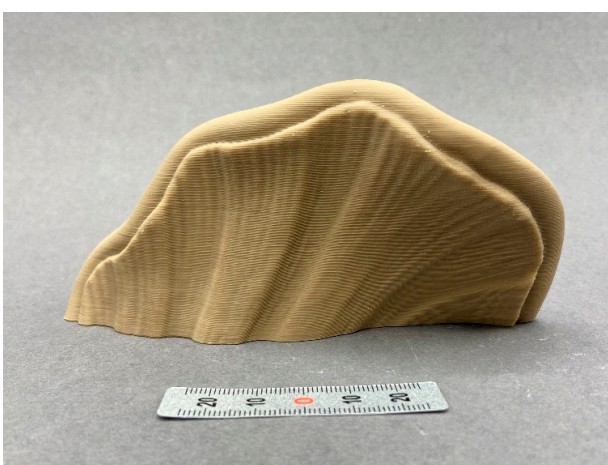

**Figure 11.** Green body of a porcelain fragment (scale bar in millimeters).

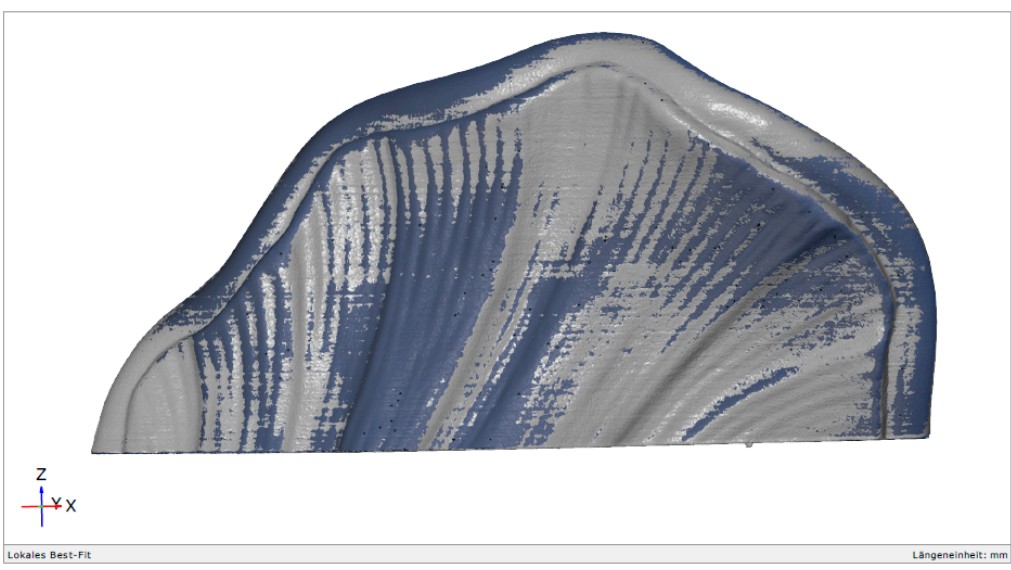

**Figure 12.** Superimposed fragment 1: CAD-model blue, green body gray; manufacturing direction: clockwise.

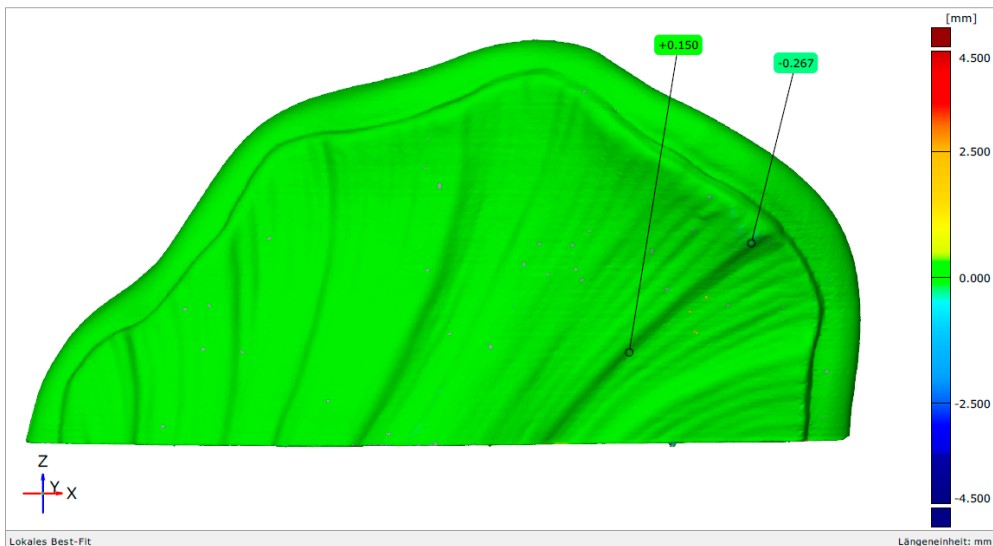

**Figure 13.** Superimposed fragment 1: manufacturing direction: clockwise; green body.

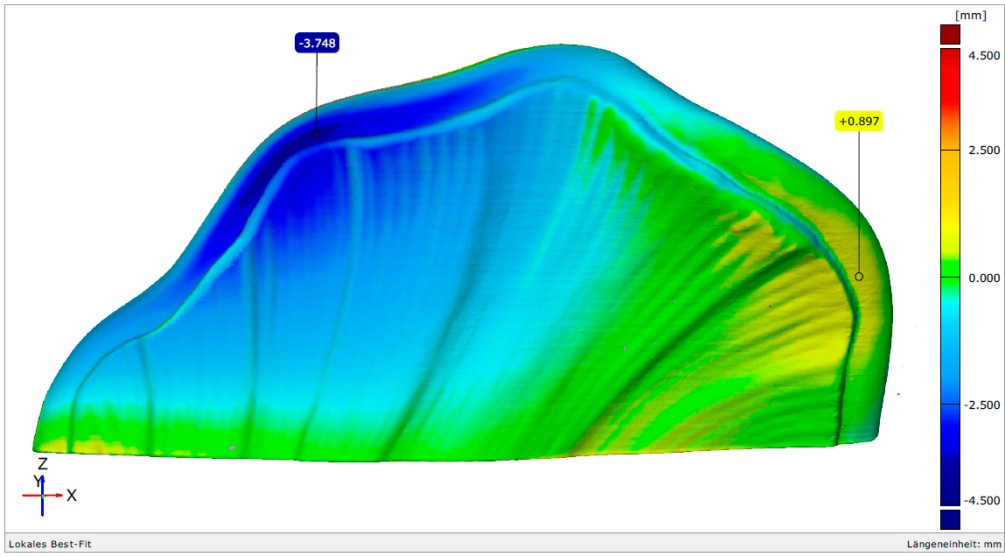

**Figure 14.** Superimposed fragment 1: manufacturing direction: clockwise; solvent debinded.

A different picture was shown when the alternated manufactured part was investigated, as shown in Figure 15. The maximum deviation is approximately ± 1 mm. This confirms the effect of the change in manufacturing direction and the relaxation during the solvent debinding in acetone. Although the mode of deformation is not a classical twisting, such as that observed during the investigation of the squares, the qualitative distortion is systematic when comparing several parts.

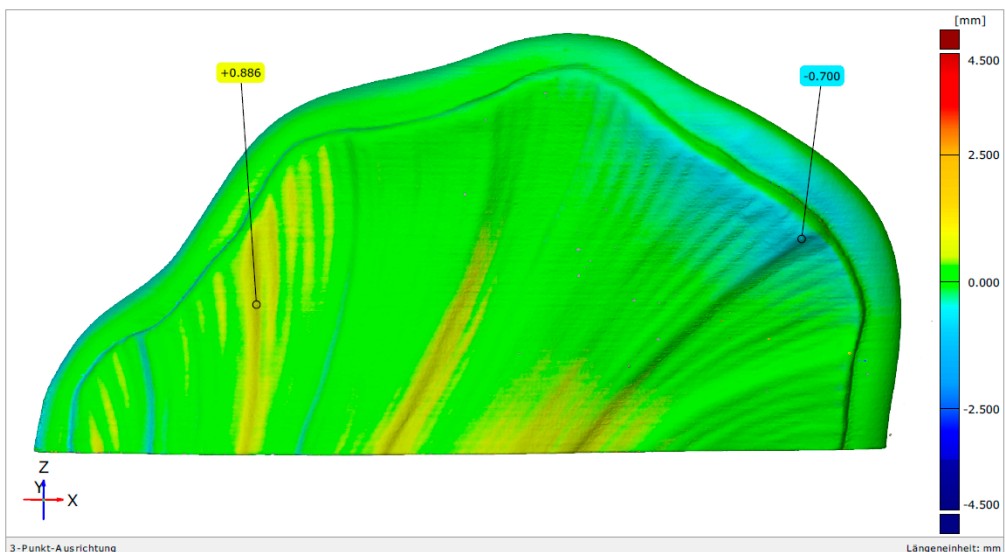

**Figure 15.** Superimposed fragment 3: manufacturing direction: alternating (*n* = 2); solvent debinded.

After sintering the clockwise manufactured part, shown in Figure 16, cracks occurred which were inter- and intra-laminar. The deformation is expectedly very high. In addition, the twisting of the baseplate (z = 0) is now clearly visible. The crack formation is caused by critical stresses during processing. The process step in which the cracks occurred is unknown, however, as the stresses and strength of the component vary in the certain processes (during solvent debinding, during thermal debinding, during sintering). In particular, during solvent debinding where the component swells [15], cracks close after drying due to the contraction of the component. Therefore, they are difficult to observe, however, they are present. They can be reopened during thermal debinding or during sintering. By applying an alternating layer deposition with *n* = 2, a completely different distortion behavior occurs. The sintered fragments exhibit a comparatively low-level distortion, of up to 1 mm, compared to the CAD-model, which is shown in Figure 17. The twisting of the baseplate is no longer visible. Despite the fact that the distortion could not be fully prevented, the influence of the alternating printing strategy is very significant.

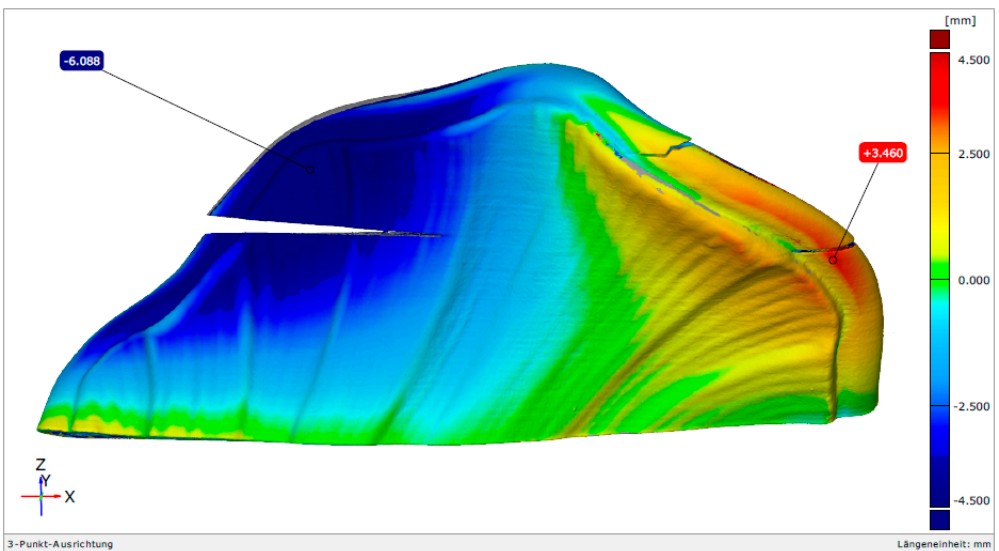

**Figure 16.** Superimposed fragment 1: manufacturing direction: clockwise; sintered.

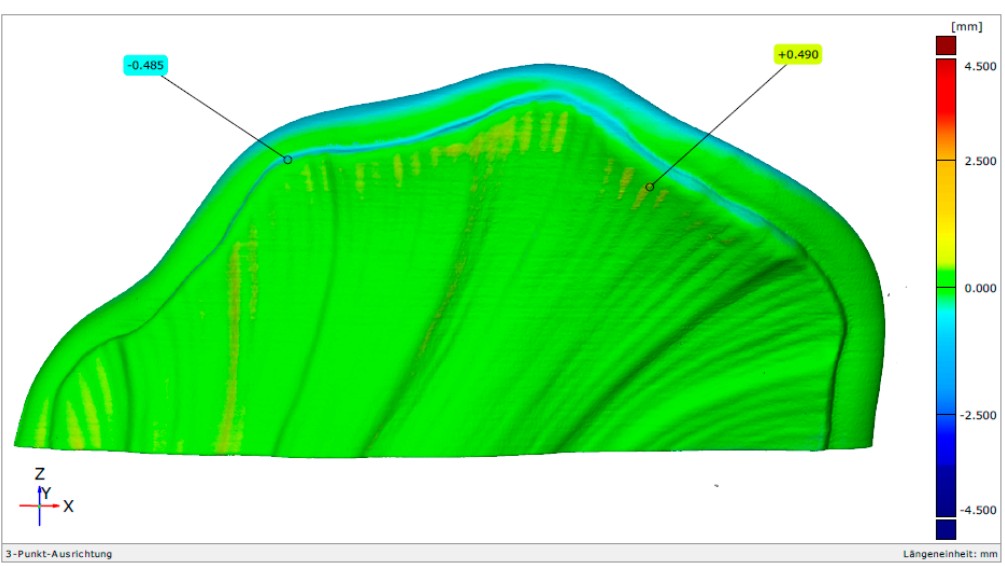

**Figure 17.** Superimposed fragment 3: manufacturing direction: alternating (*n* = 2); sintered.

After firing, a sintered density of 2.45 g/cm$^3$ was reached, indicating a low level of porosity in the range of 2 and 3% vol.

A photograph of both sintered parts is shown in Figure 18, where the gap between the baseplate and the table is clearly visible on fragment 1, which was manufactured clockwise in comparison to fragment 3, which was manufactured alternating.

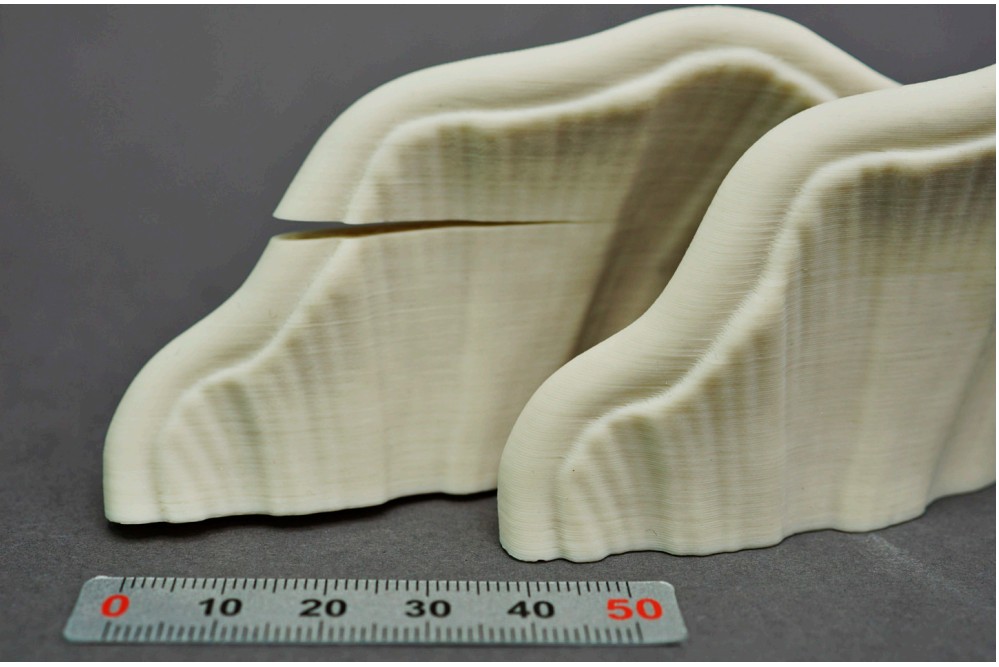

**Figure 18.** Comparison of sintered fragments; left: fragment 1 with clearly twisted base plate and crack; right: fragment 3 without twisting.

After glazing the fragments at 1250 °C with a transparent glossy glaze, the surface became very smooth and gave the part the typical appearance of Meissner porcelain, which is shown in Figure 19.

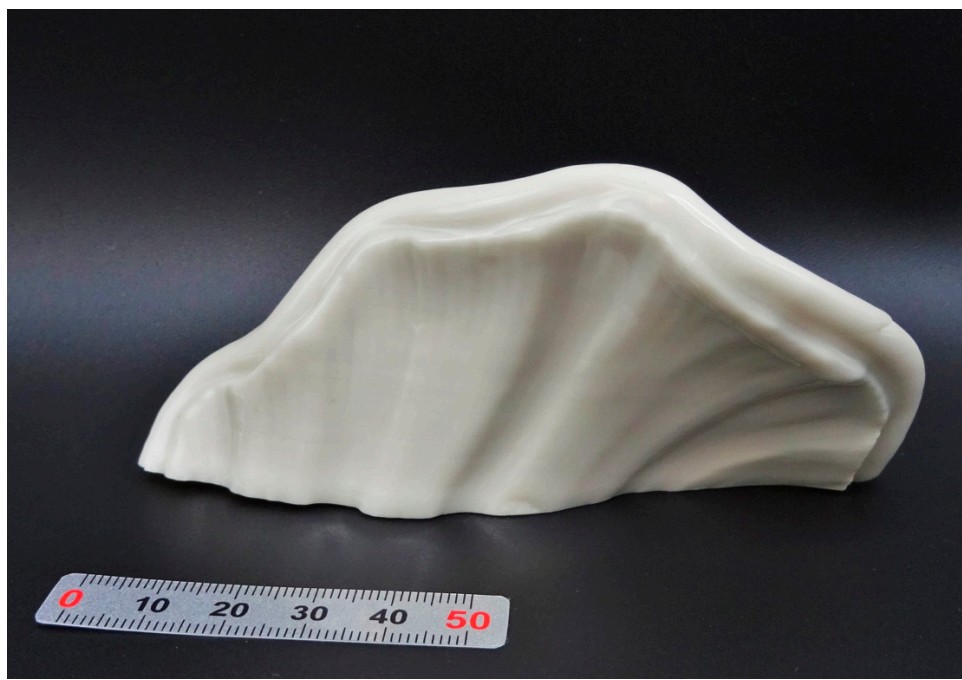

**Figure 19.** Sintered and glazed fragment, clockwise manufactured, the twisted baseplate is clearly visible in the left corner.

## 4. Conclusions and Outlook

An often-described problem in powder metallurgical manufacturing is the part distortion during processing. In particular, debinding and sintering can cause a lot of warpage, or even the formation of cracks, which leads to unsatisfying results. It was proven that the printing direction has a strong influence on the part distortion. The assumed reason for this phenomenon is the storing of stresses during the cooling of the layers, but this hypothesis has not yet been proven. Further investigations are necessary. In MEX-based processes, such as FFF (Fused Filament Fabrication) or granulate direct printing, a sacrificial polymeric binder (mostly thermoplastic) is incorporated which exhibits viscoelastic behavior depending on its rheological properties. Thus, the organics play a major role in thermoplastic MEX. The storing of stresses during shaping can lead to deformation through the relaxation of such stresses in a process step where the yield strength is lower than the present stress. The comparison between clockwise, counterclockwise, and alternating ($n = 2$) layer deposition as a first trivial approach was investigated by post-processing a standard G-code with a Python™ script. In this work, it was found that the main deformation of the used system(s) takes place in the solvent pre-debinding in acetone, in which the samples become soft. By utilizing an alternating printing, with $n = 2$ (one layer clockwise the next layer counterclockwise), the distortion of all the samples were, compared to the other uniform directional manufactured samples, very low. This behavior can also be observed using different solids and the same binder system (Figure S1) and with third party binders with different solids (Figures S2 and S3). These results encourage closer investigations of customized manufacturing directions to fully prevent distortion, reducing cracking or to specifically utilize internal stresses to distort the part in a specific way ("4D printing"), or to compensate inhomogeneous temperature gradients in the furnace. For these ideas, simulations can be very helpful to predict the stress distribution (and distortion) based on parametric models which, to the authors' knowledge, have not yet been developed. Tailoring the printing direction for perimeter and infill is not broadly implemented in standard slicing software as this issue seems not to be the priority in polymer AM technology, where the technology was established at first. Recently, Ultimaker implemented an alternating wall in version Ultimaker Cura 5.0 beta with an Arachne engine, which seems to achieve positive results with low distortion in metal printing [16]. To the best of the

authors' knowledge, no further investigation has been published. This result should draw more attention from software developers for MEX writing tailored algorithms once there is a deeper understanding of it.

For a complete restoration of the splendid Meissner vase (Figure 1), it is necessary for the upper part of the fragment to be produced without deformation and joined with a separate part, containing the fracture surface. The glazed part must then be glued on the vase by using special UV-stable organic or inorganic compounds. As the used porcelain powder is the original base material, optical and haptical properties should similarly resist all environmental factors, in contrast to polymeric substitutes used in restoration today.

**Supplementary Materials:** The following supporting information can be downloaded at: https://www.mdpi.com/article/10.3390/ceramics5040087/s1. By changing the number of layers where the manufacturing direction is changed (n), the influence of the stresses on the distortion is clearly visible, as shown in Figure S1. Here, the same binder system as used in Section 2 had been utilized for alumina filaments. The twisting decreases when an alternating printing with $n = 2$ (Figure S1 right) is applied in comparison to $n = 3$ (Figure S1 middle). Figure S1: Left: counterclockwise; middle: $n = 3$, right: $n = 2$ solvent debindered $Al_2O_3$ hexagons, scale: 1 Euro cent (pictures courtesy: 3D Ceram Sinto Tiwari GmbH). The $Al_2O_3$ (Figure S2) and the Stainless Steel 17–4 PH part (Figure S3) made with a third-party filament shows the same effect after processing. It can therefore be assumed that other binders than those used in our investigation also exhibit similar effects, leading to geometrical mismatch. Figure S2: Deformed sintered alumina part, 2.5 mm wall thickness, 45 mm height (pictures courtesy: 3D Ceram Sinto Tiwari GmbH). Figure S3: Left: sintered and twisted 17–4 PH Stainless Steel part; right: deformed sintered Stainless Steel 17–4 PH part (design courtesy: Space Structures GmbH, pictures courtesy: 3D Ceram Sinto Tiwari GmbH).

**Author Contributions:** Following contributions were made by the authors: conceptualization, J.A. and S.T.; methodology, J.A., M.R, S.T. and M.K.; software, S.T. and M.K.; validation, M.R., J.A. and U.S.; investigation, M.R., S.T. and M.K.; data curation, J.A., M.R., S.T. and M.K.; writing—original draft preparation, J.A, S.T. and M.K.; writing—review and editing, J.A., S.T., M.K., M.R. and U.S.; visualization, M.R., M.K. and J.A.; project administration, U.S. and J.A. All authors have read and agreed to the published version of the manuscript.

**Funding:** This project, entitled "RestaurAM", was funded by the German Zentrales Innovationsprogramm Mittelstand (ZIM) with the grant no. ZF4076454AG9. All authors appreciate the funding.

**Institutional Review Board Statement:** Not applicable.

**Informed Consent Statement:** Not applicable.

**Data Availability Statement:** All relevant data to follow the content are shared in the paper.

**Acknowledgments:** We acknowledge the contributions within the project, which led to new ideas and progress, by the following persons: Eric Schwarzer-Fischer (Fraunhofer IKTS, Dresden, Germany), Nadine Lorenz (Fraunhofer IKTS, Dresden, Germany), Tassilo Moritz (Fraunhofer IKTS, Dresden, Germany), Heike Ulbricht (Staatliche Kunstsammlungen, Dresden, Germany), Tobias Hackbeil (Cox3D, Pirna, Germany), Michael Bormann (KI Keramik-Institut GmbH, Meißen, Germany) and Jens Petzold (KI Keramik-Institut GmbH, Meißen, Germany).

**Conflicts of Interest:** The authors declare no conflict of interest. The funders had no role in the design of the study; in the collection, analyses, or interpretation of data; in the writing of the manuscript or in the decision to publish the results.

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
