# Peer review of "Reducing the Distortion in Particle Filled Material Extrusion (MEX)-Based Additive Manufacturing (AM) by Means of Modifying the Printing Strategy"

_ceramics, doi:10.3390/ceramics5040087_

Round 1
Reviewer 1 Report
1. Advantages and disadvantages of the MEX method based on FFF.
2. What are the potential applications for other materials?
3. Besides the Cer-AM-FFF method, Is there any other method to minimize the residual stress causing deformation to the material?
Author Response
Dear reviewer,
thank you for your comments and interest on our works. We highly appreciate your qualified input.
All my best.

Reviewer 2 Report
The paper is reporting a strategy to reduce the distortion in CerAM FFF processed samples, that is, tuning the printing direction and its combination. This result is interesting, but the work lacks deep understaning on the phenomena, in other words, why these stuff introduce such difference in stress , as mentioned by the authors. The reviewer suggests the authors give more space to discuss this issue. Besides, there are some flaws and typos in current version. For example, Figure 5 right is missing. The scale bar in Figure 1 left is missing. Line 112-115, line 263, line 271
Author Response

(The authors gave the same response as above.)

Reviewer 3 Report
The authors proposed the approach through changing the use of thermoplastic porcelain filaments with the solid loading to reduce residual stresses in components produced by CerAM FFF. This work seems interesting and can have a good effect on reducing deformation of ceramics and metals in additive manufacturing process.
1. Avoid abbreviations in Keywords section.
2. There is an error in the numbering of the secondary title in the paper, such as "3. Results and discussion" and "3. Other examples:" Please check the full text carefully!
3. It is excessively cumbersome for a research article with more than 20 pictures. Important data pictures or images that can give important information can be placed in the manuscript, and it is recommended that other pictures can be placed in the supplementary material.
4. Similarly, it is also recommended to put some parts of describing the experimental process or operation into supplementary material to keep the article scientific and refined.
5. The approach proposed by the authors to solve the deformation problem has good universality. In Other examples section, the content of this section is too simple and short. It is suggested that this part be supplemented and described in detail.
6. In the Conclusion section, it is suggested that the authors should give a general regular summary of the approach proposed to solve the problem of ceramic and metal deformation during additive manufacturing, which is not only a sublimation of the article, but also easier for readers to understand.
7. The references are insufficiently cited and do not involve the latest 3D printing progress, most of which are long ago. It is recommended to quote the latest literature related to research progress of 3D printing, such as (Chemical Engineering Journal, 2022, 433, 134341), (Green Chemical Engineering, 2022, 10.1016/j.gce.2022.04.005), etc.
Author Response

(The authors gave the same response as above.)

Round 2
Reviewer 2 Report
The authors has responsed to my comments
Reviewer 3 Report
The article has been properly updated, and the current version is suitable for publication.
As a minor comment, it is recommended to quote the latest literature related to the research progress of 3D printing.